# Benzothiophene Adsorptive Desulfurization onto trihexYl(tetradecyl)phosphonium Dicyanamide Ionic-Liquid-Modified Renewable Carbon: Kinetic, Equilibrium and UV Spectroscopy Investigations

**DOI:** 10.3390/molecules28010298

**Published:** 2022-12-30

**Authors:** Mohamed A. Habila, Zied A. ALOthman, Monerah R. ALOthman, Mohammed Salah El-Din Hassouna

**Affiliations:** 1Chemistry Department, College of Science, King Saud University, Riyadh 11451, Saudi Arabia; 2Plant and Microbiology Department, College of Science, King Saud University, Riyadh 11451, Saudi Arabia; 3Department of Environmental Studies, Institute of Graduate Studies and Research, Alexandria University, Alexandria 5424041, Egypt

**Keywords:** desulfurization, adsorption, renewable carbon, ionic liquids, recycling, UV spectroscopy

## Abstract

The negative environmental and industrial impacts of the presence of sulfur compounds such as benzothiophene in fuels have led to a greater interest in desulfurization research. In this work, carbon from palm waste sources was modified with trihexYl(tetradecyl)phosphonium dicyanamide-ionic liquid and characterized by SEM, EDS, XRD and FTIR to assess surface properties. Then, the prepared carbon and carbon modified with ionic liquid were evaluated for the adsorption of benzothiophene by investigating the effects of time. The equilibrium occurred after 120 min, recording adsorption capacities of 192 and 238 mg/g for carbon and carbon modified with ionic liquid, respectively. The effect of the adsorbent dose on the adsorption of benzothiophene was evaluated, indicating that the maximum adsorption capacities were obtained using a dose of 1 g/L for both carbon and carbon modified with ionic liquid. The kinetic investigation for the adsorption of benzothiophene onto carbon and carbon modified with ionic liquid indicated that the second-order kinetic model is well fitted with the adsorption data rather than the first-order kinetic model. The equilibrium investigations for the adsorption of benzothiophene onto carbon and carbon modified with ionic liquid with Langmuir and Freundlich isotherm models reveals that the Freundlich model is the most suitable for describing the adsorption process, suggesting a multilayer adsorption mechanism. The desulfurization process showed a high impact on environmental safety due to the possibility of regenerating and reusing the prepared adsorbents with promising results up to five cycles.

## 1. Introduction

At present, the desulfurization of crude oil and fuels has received great interest from scientists and researchers. The problem with petroleum oil and fuels that have a high sulfur content is that sulfur poisons the catalysts during refining processes and corrodes the pipeline, pumping and refining equipment [1,2]. Additionally, it produces undesired gases, such as SO_x_, the main air pollutant that causes greenhouse problems and acid rain. Therefore, great attention has been paid to sulfur problems; environmental regulations have placed stringent limits on the sulfur content in transportation fuels, at <10 ppm, to prevent or reduce sulfur dioxide emissions [3]. The catalytic hydrodesulfurization (HDS) method, which is usually used in refineries, is an expensive and energy-intensive process because of the high temperature and pressure needs. In addition, the need to convert most by-products into acceptable products comes at an extra cost [4,5]. Additionally, the HDS process produces high amounts of saturated olefins, which lower the octane rating of the fuel. Thus, researchers usually seek an alternative process in order to avoid the use of hydrogen gas [6,7]. Recently, desulfurization by adsorption showed a safer and high sulfur removal efficiency. In the reactive adsorption process, the main attractive force for sulfur-containing compounds depends on the bond formation between the sulfur atom in the compounds and the metal atom in the adsorbent; therefore, this force is not affected by fuel composition [8,9]. Adsorption-based desulfurization processes have advantages, such as fast separation, easy operation and cheap when applying low-cost adsorbents and highly effective desulfurization techniques [10,11,12,13,14,15].

Many previous investigations have been developed for desulfurization. For example, Choi et al. studied the removal of benzothiophene sulfone by adsorption onto clay materials, which showed an adsorption behavior fitted with the Langmuir and Freundlich models confirming monolayer adsorption and heterogeneous surfaces. In addition, the adsorption process was found to follow a second-order kinetic model [16]. Amiri et al. prepared flower-shape-like (Bi(Bi_2_S_3_)_9_I_3_)_2/3_ nanomaterials for the photocatalytic desulfurization of benzothiophene [17]. Shamsaee et al. applied a dynamic electroreduction method for the removal of benzothiophene using model diesel [9]. Zhang et al. prepared silver-modified UiO-66 to enhance the adsorptive desulfurization of benzothiophene with the possibility of regeneration four times [18]. Ferrela et al. applied a heterogeneous *cis*-dioxomolybdenum(VI)-based catalyst for the oxidation of benzothiophene and tested imidazolium-based ionic liquids to enhance extractive oxidative desulfurization. Their results showed that the presence of imidazolium ionic liquids has a dual role as an extractant and reaction medium, in addition to acting as a stabilizing agent for the oxidant and the *cis*-dioxomolybdenum(VI) catalyst [19]. Fujiiki et al. developed an adsorptive desulfurization process by applying a heat-treated silica gel to enhance the desulfurization of benzothiophene and reported that the heat treatment of silica enables the selective removal of benzothiophene [20]. Wang et al. fabricated Pr/Ce-N-TiO_2_ for the desulfurization of sulfur compounds through visible light and assessed destroying of carbon–sulfur bonds in benzothiophene [21]. Zu et al. constructed cerium species with zeolites and yttrium modification and recorded a noticeable adsorption performance for the removal of sulfur compounds, including benzothiophene, even after regeneration two times [22]. Lee et al. developed a desulfurization technique based on the adsorption onto bimetallic Cu- and Ce-doped Y zeolites for the removal of benzothiophene and dibenzothiophene in octane, benzene, and naphthalene with great enhancement compared to the unmodified Y zeolite [2].

Renewable activated carbon can be fabricated from biomass or waste sources, which enhances recycling materials and produce low-cost adsorbent [23,24,25]. Activated carbon is known as an efficient, stable and porous adsorbent that possesses a wide range of applications, including adsorption [26,27,28,29,30]. It is reported that activated carbons can be used for adsorptive desulfurization [28]. However, the adsorption efficiency of activated carbon towards sulfur is low. Therefore, the improvement of activated carbon efficiency towards sulfur adsorption is necessary. Khan et al. fabricated CuCl_2_-decorated carbon materials as adsorbent for benzothiophene and reported an enhancement of about 30% adsorption efficiency than the unmodified carbon [31]. On the other hand, ionic liquid is considered a green solvent with high stability and extraction efficiency [32,33]. Zhao et al. applied iron porphyrins together with 1-butyl-3-methylimidazolium hexafluorophosphate ([Bmim]PF_6_) for desulfurization purposes, reporting the recycling possibility of six times [5]. Zhao et al. reported the effectiveness of ionic liquids for extractive desulfurization with a promising efficiency related to the partitioning of aromatic sulfur from the solution to the ionic liquid phase [34]. Ibrahim et al. prepared ionic liquid of 1-butyl-3-methylimidazolium dicyano(nitroso)methanide ([C4mim][dcnm]), for extractive desulfurization to remove aromatic sulfur species from gasoline [35]. Kulkarni et al. reviewed ionic liquid for the desulfurization process as a green and environmentally friendly method [6]. Therefore, this work aims to modify renewable carbon from waste sources with ionic liquid and investigates its efficiency for desulfurization application by the adsorption of benzothiophene, in addition to characterizing the kinetics and isotherms of adsorption process for benzothiophene removal. Furthermore, the regeneration and the reuse of the prepared carbon-modified ionic liquid were investigated.

## 2. Results and Discussion

### 2.1. Structure and Morphological Feature of the Fabricated trihexYl(tetradecyl)phosphonium Dicyanamide-Modified Renewable Carbon

The crystalline structures of carbon and the trihexYl(tetradecyl)phosphonium dicyanamide-modified renewable carbon were characterized by XRD diffraction, as presented in Figure 1A,B. Peaks at 2θ of 23 were detected for carbon and trihexYl(tetradecyl)phosphonium dicyanamide-modified renewable carbon due to the amorphous structure (Figure 1A,B), confirming that the modification of carbon with ionic liquid keeps the original carbon structure. The unmodified carbon showed peaks at 2θ of 32, 35, 37 and 68 (Figure 1A), which may be attributed to impurities species due to the carbon originating from waste sources as well as being activated with zinc chloride. The impurity-related peaks disappeared in the trihexYl(tetradecyl)phosphonium dicyanamide-modified renewable carbon due to shielding. The detection of XRD peaks related to impurities associated with activated carbon was previously reported by Osman et al. [36]. In addition, the surface functional groups for carbon and trihexYl(tetradecyl)phosphonium dicyanamide-modified renewable carbon were detected by FTIR (Figure 2A,B). Both the original renewable carbon and trihexYl(tetradecyl)phosphonium dicyanamide-modified renewable carbon showed the presence of C=C, which appears in the range of 1500–1580 cm^−1^. In addition, the trihexYl(tetradecyl)phosphonium dicyanamide-modified renewable carbon exhibited rich C-H stretching of aliphatic structure with clear peaks at 2800 cm^−1^. The peak at 1459 cm^−1^ is related to P-CH_2_-CH_3_, while the peak at 717 cm^−1^ is due to the P-C bond. The peaks at 800 cm^−1^ and 1361 cm^−1^ are due to the C-H bending of ionic liquid. The peak at 1027 cm^−1^ may be attributed to the C-N of trihexYl(tetradecyl)phosphonium dicyanamide-modified renewable carbon. These detected surface groups confirm the formation of an ionic liquid layer on the carbon surfaces. The post-carbon modification enables the design of task-specific materials for targeted adsorbate. This modification allows the combination of various materials with different physicochemical characteristics, which results in the improvement of stability and surface functionality to enhance the oriented application. It has been reported that ionic liquids exhibit highly stable and surface-rich functional groups due to the presence of multi-cation–anion structure combinations that enhance the materials’ stability. These promised characteristics are expected to enhance the targeted adsorptive desulfurization presented in this work to remove benzothiophene.

The prepared adsorbents were examined for morphology characterization by SEM, as presented in Figure 3. The renewable activated carbon from waste sources showed a rough surface that included cavities and a perforated structure with internal tubular pores (Figure 3A,B). The surface elemental analysis from the EDS examination showed that the main elements are carbon and oxygen (Figure 3C). The trihexYl(tetradecyl)phosphonium dicyanamide-modified renewable carbon showed a similar morphology to the original carbon but without the appearance of pores due to their occupation with ionic liquid molecules (Figure 3D,E). The surface elemental analysis confirmed that the main elements were carbon and oxygen in the case of the original carbon (Figure 3C), while for trihexYl(tetradecyl) phosphonium dicyanamide-modified renewable carbon, the main surface elements were carbon, oxygen and phosphorous. The BET surface area of the original carbon was 163.8 m^2^/g, while for trihexYl(tetradecyl)phosphonium dicyanamide-modified renewable carbon, it was 5.1 m^2^/g, indicating the noticeable decrease in the porous character during modification. These results agree with those reported by Fatima et al. for rubber-seed-shell-derived activated carbon and rubber-seed-shell-derived activated carbon modified with [bmpy][Tf2N] ionic liquid (IL), which had a surface area of 683 m^2^/g and 14 m^2^/g, respectively [37]. In addition, Yusuf et al. prepared ionic-liquid-impregnated carbon materials and reported a surface area of 863 and 117 for the activated carbon and impregnated carbon, respectively [38]. The impregnation-based modification of adsorbent depends on filling the pores and staking the modifier on the original materials by electrostatic interaction, including interactions of Van der Waal forces. As a result, the surface area is reduced in the final modified materials. The incorporation of ionic liquid of trihexYl(tetradecyl) phosphonium dicyanamide onto the carbon surface introduced the P element to the adsorbent surface, which enhances the interaction with benzothiophene and improves the adsorption capacity.

### 2.2. Adsorption of Benzothiophene Using Carbon and trihexYl(tetradecyl)phosphonium Dicyanamide-Modified Renewable Carbon

Carbon and the carbon modified with ionic liquid were evaluated for the adsorption of benzothiophene. By investigating the effects of time ranging from 1 to 280 min, the equilibrium state occurred after 120 min, recording adsorption capacities of 192 and 238 mg/g for carbon and carbon modified with ionic liquid, respectively (Figure 4A). After 120 min, there was no further increase in the adsorption capacity of benzothiophene due to the steady state and surface saturation. In this case, the adoptive desulfurization exhibited a high capacity to remove benzothiophene. In addition, a noticeable improvement was achieved by ionic liquid modification, as even the surface area was lower than that of the original carbon. This may be attributed to the presence of surface functional groups, such as P-CH_2_-CH_3_, P-C, C-H, C=C and C-N, in the trihexYl(tetradecyl)phosphonium dicyanamide-modified renewable carbon, which act as driving forces for attracting adsorbate benzothiophene compounds during adsorptive desulfurization process. However, the adsorptive desulfurization for the removal of benzothiophene in the case of original carbon can be attributed to the surface area enhancement together with the electro-optic interaction between the carbon structure and their C=C bonds and benzothiophene, which may include interactions of Van der Waals forces and π–π interaction.

The effect of the adsorbent dose on the adsorption of benzothiophene was evaluated as presented in Figure 4B, indicating that the maximum adsorption capacities were obtained using a dose of 1 g/L for both carbon and carbon modified with ionic liquid. It has been reported that the lower adsorbent dose leads to a higher adsorption capacity due to the complete usage of all the adsorbent surfaces.

The kinetic investigation for the adsorption of benzothiophene onto carbon and carbon modified with ionic liquid was evaluated by applying pseudo-first-order (Equation (1)) and pseudo-second-order kinetic (Equation (2)) models.
log(q_e_ − q_t_) = log q_e_ − K_1_t/2.303(1)

Figure 5A presents the plot of log(q_e_ − q_t_) and t, and from the slope and intercept, the values of k1 and qe were calculated (Table 1).
t/q_t_ = 1/Kq_e_^2^ + 1/q_e_ t(2)

Figure 5B shows the plot of t/q_t_ and t, from which q_e_ and k can be determined for the adsorption of carbon and trihexYl(tetradecyl)phosphonium dicyanamide-modified renewable carbon.

The obtained results for the kinetic constant presented in Table 1 indicate a poor agreement between the values of experimental q and calculated q_e_ in the case of the pseudo-first-order kinetic model. However, the values of experimental q and calculated q showed a good correlation in the case of applying the pseudo-second-order kinetic model, revealing that the pseudo-second-order assumption is the most consistent for describing the adsorption of carbon and trihexYl(tetradecyl)phosphonium dicyanamide-modified renewable carbon.

### 2.3. Isotherm Studies

#### 2.3.1. Langmuir Isotherm

The Langmuir Equation (3) was applied for the adsorption data of benzothiophene onto carbon and trihexYl(tetradecyl)phosphonium dicyanamide-modified renewable carbon to assess the adsorption layers during the desulfurization process:(3)$$\frac{C_{e}}{q_{e}}=(\frac{1}{Q_{max}^{o}})\;C_{e}+\frac{1}{Q_{max}^{o}\;K_{L}}\;$$
where *Q^o^_max_* (mg/g) is the maximum adsorption capacity for the adsorption of benzothiophene onto carbon and trihexYl(tetradecyl)phosphonium dicyanamide-modified renewable carbon, *C_e_* (mg/L) is the benzothiophene concentration at equilibrium, *q_e_* (mg/g) is the amount of benzothiophene uptake at equilibrium, and *K_L_* (L/mg) is a constant related to the affinity between benzothiophene and carbon or trihexYl(tetradecyl)phosphonium dicyanamide-modified renewable carbon.

Figure 6A shows the application of Langmuir’s assumption to the adsorption data of benzothiophene onto carbon and carbon@ionic liquid. The correlation coefficient, *R*^2^, is weak, confirming that the Langmuir model is not compatible with describing the obtained adsorption data.

#### 2.3.2. Freundlich Isotherm

The Freundlich isotherm, in Equation (4), was utilized to correlate the adsorption data of benzothiophene onto carbon and carbon@ionic liquid to assess the adsorption layers during the desulfurization process:Logqe = logK + 1/n logCe(4)
where qe is the amount of benzothiophene uptake in mg/g at equilibrium onto carbon@ionic liquid, Ce (mg/L) is the benzothiophene concentration at equilibrium, K_F_ (mg/g)/(mg/L)^n^ is the Freundlich constant, and n is the Freundlich intensity parameter, related to the magnitude of the adsorption driving force or the surface heterogeneity.

The Freundlich constants (K and n) was calculated by plotting logqe versus logCe (Figure 6B). In addition, the obtained results from applying the Freundlich model showed the linear form (Table 2 shows the calculated constants). The obtained qmax, in the case of the Langmuir and Freundlich isotherms, and the correlation coefficients indicated that the Freundlich isotherms are the most suitable for describing the adsorption of benzothiophene onto carbon and carbon@ionic liquid during the desulfurization process. The results suggest a multi-adsorption layer.

### 2.4. Sustainability and Regeneration Approach

The regeneration and reuse of adsorbent materials are the most important issues in the context of environmental safety [39,40]. The effectiveness of the regeneration process depends on the ability to control the desorption process and clean the adsorbent surface in order to be ready for subsequent use. The results show a high regeneration efficiency up to the fifth use, which reveals the effectiveness of the prepared adsorbents for the multi-use for desulfurization process by benzothiophene adsorption (Figure 7). Recycling adsorbents for desulfurization purposes have been reported to enhance the entire treatment process and reduce environmental pollution. For example, Wang et al. used Schiff base–metal complexes for the desulfurization of dibenzothiophene with an adsorption capacity of 21.66 mg/g and the possibility of reusing without an adsorption capacity reduction of five times [41]. Qiu et al. modified coal tar to fabricate carbon as an adsorbent for benzothiophene and reported an adsorption capacity of 32.8 mg g^−1^. The prepared carbon can be easily recycled and regenerated, achieving an efficiency of 98%, 95%, and 91% of the initial adsorption capacity after the first, second, and third uses, respectively [42].

The achieved adsorptive desulfurization obtained in this work for the removal of benzothiophene onto activated carbon and trihexYl(tetradecyl)phosphonium dicyanamide-modified renewable carbon was compared with other adsorbents from the literature (Table 3). The adsorptive-based desulfurization process using HKUST-1 for the removal of benzothiophene, reported by Qiu et al., exhibits an adsorption capacity of 14.4 mg/g, which is lower than both the activated carbon as well as trihexYl(tetradecyl)phosphonium dicyanamide-modified renewable carbon [43]. Saleh et al. reported a maximum adsorption capacity of 5.7 mg g^−1^ for the modification of activated carbon with manganese oxide to enhance the adsorptive capacity [15]. In addition, the application of coal-tar-derived carbon as an adsorbent for benzothiophene showed an adsorption capacity of 32.8 mg/g [42]. The comparison confirms the superior capacity of the activated carbon and trihexYl(tetradecyl)phosphonium dicyanamide-modified renewable carbon for the removal of benzothiophene and suggests the effectiveness of the ionic liquid modification, which significantly activates the surface towards benzothiophene uptake.

## 3. Materials and Methods

### 3.1. Preparation of trihexYl(tetradecyl)phosphonium Bis2,4,4-(trimethylpentyl)phosphinate (Cyphos^®^ IL 104)-Modified Activated Carbon

The chemicals, solvents and reagents applied in this research were of high purity. trihexYl(tetradecyl)phosphonium dicyanamide (CYPHOS^®^ IL 105) was obtained from Cytec Industries Inc., Woodland Park, NJ 07424, USA. Renewable carbon was prepared from palm waste, as described in our previous work [44]. Heptane, octanol, acetic acid and benzothiophene were purchased from Sigma, St. Louis, MO, USA. The reagents were diluted day by day to obtain working solutions. The modification of the activated carbon from mixed recyclable waste was conducted by impregnation as described in the literature with some modifications [45]. In detail, 3 grams of the renewable carbon from waste sources were mixed with 100 mL of 0.25% (*m*/*v*) trihexYl(tetradecyl) phosphonium Bis2,4,4-(trimethylpentyl)phosphinate (Cyphos^®^ IL 104) in acetone medium, and the mixture was continuously stirred for 24 h. Then, the formed carbon@ionic liquid was separated by filtration, washed with deionized water and dried at 105 °C for 24 h. The original carbon, as well as the fabricated trihexYl(tetradecyl)phosphonium Bis2,4,4-(trimethylpentyl)phosphinate (Cyphos^®^ IL 104)-modified activated carbon were characterized by XRD (X’Pert PRO MPD, PANalytical, Almelo, The Netherlands), FTIR (Vertex-80 spectrometer Bruker, Billerica, MA, USA), SEM and EDS (JSM-7600F, JEOL, Tokyo, Japan) and surface area analysis (Micromeritics, Gemini VII, 2390 Surface Area and Porosity USA).

### 3.2. Application of the Carbon and Carbon@ionic Liquid for the Adsorptive Desulfurization of Boenzothiophene

The adsorptive-based desulfurization of benzothiophene was applied using batch experiments. The adsorbent materials, including carbon and carbon@ionic liquid, were mixed separately with the benzothiophene solution in heptane and shacked for a certain time at 150 rpm. Then, the phases were separated by centrifugation. The adsorbed benzothiophene concentration was evaluated by UV-visible spectroscopy (Thermo Scientific, Cleveland, OH, USA) for the detection of the benzothiophene concentration at a wavelength of 291 nm before and after the adsorption process. The same procedures were repeated to investigate the effect of contact time, adsorbent dose and benzothiophene concentration. Blank experiments were conducted in parallel to the experiments. The adsorptive desulfurization capacity for the carbon and carbon@ionic liquid was calculated using Equation (5):(5)$$q_{e}=\frac{\left(C_{0}-C_{e\;}\right)*V}{M}$$
where *C*_0_ represents the initial benzothiophene concentration, *C_e_* is the final benzothiophene concentration after the adsorption process, *V* is the volume, and *M* is the adsorbent mass for carbon or carbon@ionic liquid in grams.

For the regeneration of the carbon and carbon@ionic liquid adsorbents, the process described by Liu et al. [46] was followed with some modifications. In detail, the regeneration solution was prepared by mixing octanol and acetic acid with a ratio of 9:1. After each adsorption cycle, the carbon and the carbon@ionic liquid were subjected to washing with octanol/acetic acid (9:1) mixture and dried at 105 °C before the subsequent use. The adsorption/desorption regeneration study for the removal of benzothiophene was repeated for five investigations.

## 4. Conclusions

The modification of carbon with ionic liquid led to an improvement in the adsorption capacity for the removal of benzothiophene. The most suitable contact time to achieve the maximum adsorption capacities of 192 and 238 mg/g for carbon and carbon@ionic liquid, respectively, is 120 min. The adsorptive desulfurization process follows the second-order kinetic model and freindlish isotherm for both carbon and ionic liquid-modified carbon, suggesting heterogeneous surfaces and the occurrence of the multilayer adsorption process. The modification of the carbon surface investigated in the present study presenting positive results for the enhancement of adsorptive desulfurization will lead to future research addressing the treatment of carbon surfaces with various ligands and/or metal oxides to assess their effects on adsorption efficiency. In addition, the research in the future can be extended to carbon modified with ionic liquid-based materials for the adsorption of various environmental pollutants, such as heavy metals, dyes and other sulfur compounds, including thiophene and dibenzothiophene.

## Figures and Tables

**Figure 1 molecules-28-00298-f001:**
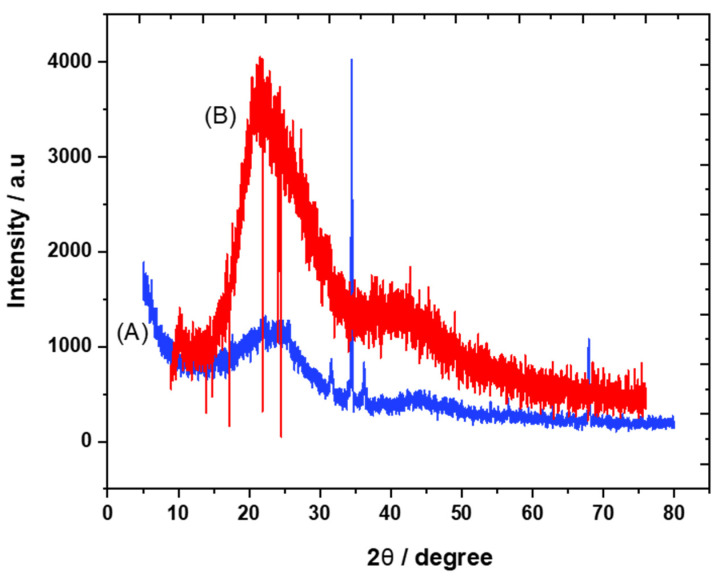
XRD pattern of carbon (**A**) and trihexYl(tetradecyl)phosphonium dicyanamide-modified renewable carbon (**B**).

**Figure 2 molecules-28-00298-f002:**
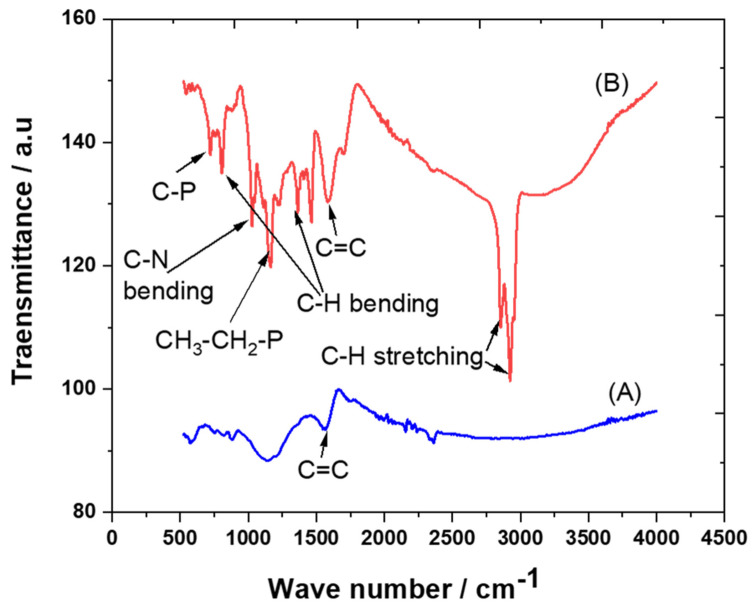
FTIR spectra of carbon (**A**) and trihexYl(tetradecyl)phosphonium dicyanamide-modified renewable carbon (**B**).

**Figure 3 molecules-28-00298-f003:**
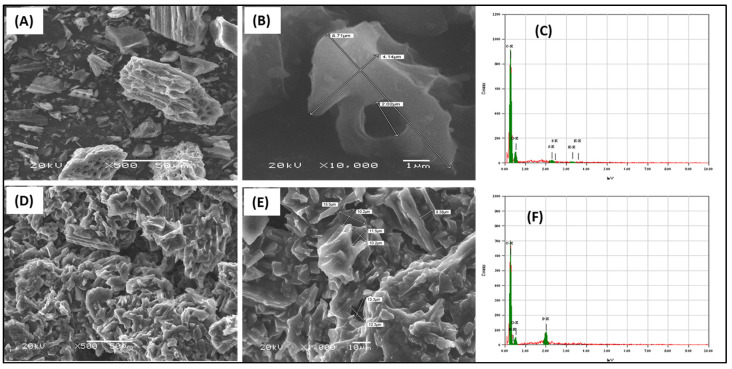
SEM images of carbon (**A**,**B**) and trihexYl(tetradecyl)phosphonium dicyanamide-modified renewable carbon (**D**,**E**) and the EDS of carbon (**C**) and trihexYl(tetradecyl)phosphonium dicyanamide-modified renewable carbon (**F**).

**Figure 4 molecules-28-00298-f004:**
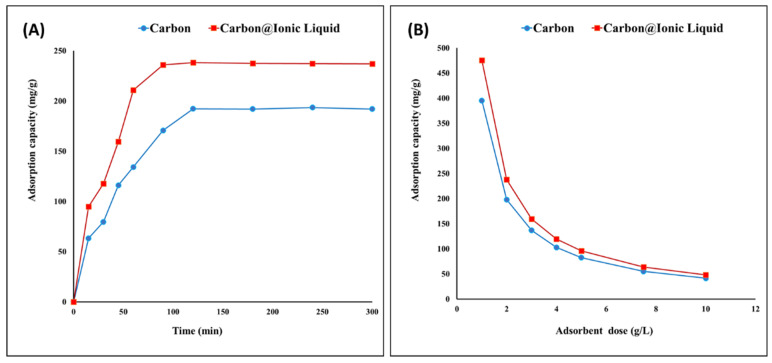
Adsorption of benzothiophene onto carbon and trihexYl(tetradecyl)phosphonium dicyanamide-modified renewable carbon: effect of contact time (**A**) and effect of adsorbent dose (**B**).

**Figure 5 molecules-28-00298-f005:**
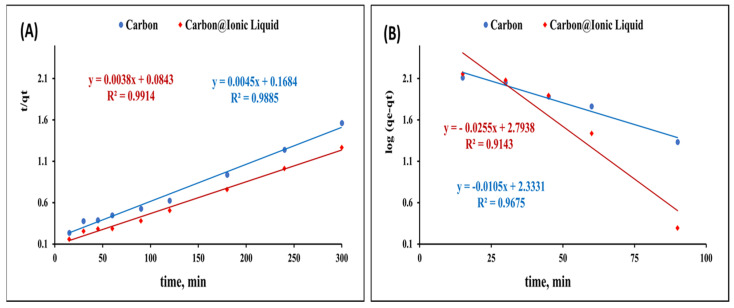
Kinetic investigation for the adsorption of benzothiophene onto carbon and trihexYl(tetradecyl)phosphonium dicyanamide-modified renewable carbon: second-order kinetic model (**A**) and first-order kinetic model (**B**).

**Figure 6 molecules-28-00298-f006:**
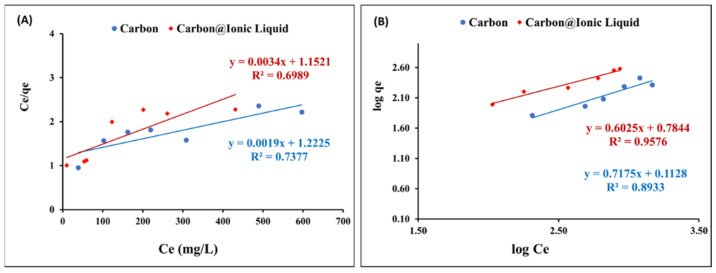
Equilibrium investigation for the adsorption of benzothiophene onto carbon and carbon@ionic liquid: Langmuir (**A**) and Freundlich (**B**) isotherms.

**Figure 7 molecules-28-00298-f007:**
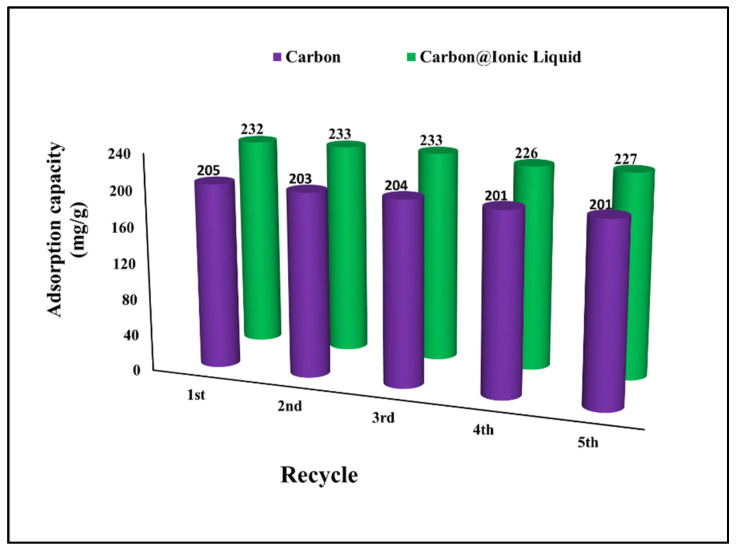
Investigation of regeneration and reusing for the adsorption of benzothiophene onto carbon and carbon@ionic liquid.

**Table 1 molecules-28-00298-t001:** Kinetic investigation for the adsorption of benzothiophene onto carbon and trihexYl(tetradecyl)phosphonium dicyanamide-modified renewable carbon.

		Pseudo-First Order			Pseudo-Second Order		
	qe,exp (mg/g)	K_1_ (min^−1^)	qe,cal (mg/g)	*R* ^2^	K_2_ (g/mg·min)	qe,cal (mg/g)	*R* ^2^
**Carbon**	192	0.024	215	96	1.2 × 10^−4^	222.2	98
**Carbon@ionic liquid**	238	0.058	622	91	1.7 × 10^−4^	263.1	99

**Table 2 molecules-28-00298-t002:** Langmuir and Freundlich constants for the adsorption of benzothiophene onto carbon and carbon@ionic liquid.

	Langmuir Constants			Freundlich Constants		
	K_L_	Q_max_	*R* ^2^	K_F_	n	*R* ^2^
**Carbon**	1.55	526	73	1.29	1.39	89
**Carbon@ionic liquid**	2.95	294	69	6.08	1.65	95

**Table 3 molecules-28-00298-t003:** Comparison of the adsorption capacities for the removal of benzothiophene onto activated carbon and trihexYl(tetradecyl)phosphonium dicyanamide-modified renewable carbon with those of other adsorbents.

Adsorbent	Adsorption Capacity (mg/g)	References
HKUST-1	14.4	[43]
Activated carbon modified with manganese oxide	5.7	[15]
Coal-tar-derived carbon	32.8	[42]
Activated carbon	192	This work
trihexYl(tetradecyl)phosphonium dicyanamide-modified renewable carbon	238	This work

## Data Availability

The data presented in this study are available on request from the corresponding author.

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
