# Peer review of "Benzothiophene Adsorptive Desulfurization onto trihexYl(tetradecyl)phosphonium Dicyanamide Ionic-Liquid-Modified Renewable Carbon: Kinetic, Equilibrium and UV Spectroscopy Investigations"

_molecules, 2022, doi:10.3390/molecules28010298_

Round 1
Reviewer 1 Report
This paper sutdied the benzothiophene adsorptive desulfurization onto Tri- hexyl(tetradecyl)phosphonium dicyanamide ionic liquid modi-fied renewable carbon. After having a careful reading of this paper, I believe the following issues must be resovled before its publication.
1. In the introduction section, the authors should avoid too much direct listing of the results of the literature. It is important to highlight the more relevant literature and to make more summary statements about the literature.
2. There are more spurious peaks in the XRD pattern of this paper, not suitable for the interpretation of the pattern results. What’s more, it is recommended to supplement the BET results for the deep analysis of this material.
3. The changes of functional groups should be highlighted in the FT-IR spectra results. What’s more, the understanding of "Confirming the formation of ionic liquid layer on the carbon surfaces" is not reasonable, while the authors should give more direct evidence.
4. The EDS analysis result did not reveal the elemental changes of the material after modification, for example the increase of P element in the material after modification.
5. The adsorption process should be further analyzed , based on the SEM, BET, and other related characterization results.
6. In this paper, the illustration for the relationship between the stucture of carbon material and its adsorption performance is insufficient.
7. I am wondering whether the auhtors have studiedthe effect of different concentrations of ionic liquids on the modification of the materials?
In addition, some word spelling errors, uniformity of unit writing format, and consistency of material names (abbreviations are recommended) should be noted.
Author Response
Thanks for your time and your efforts to review our manuscript. Your valuable comments will improve the final version of the manuscript. We have considered your comments and correct the manuscript according to it, as in the attached response to reviewer file. All changes are marked with red color font.

Reviewer 2 Report
This work is devoted to the removal of benzothiophene from heptane solutions using bioadsorbents. The topic is interesting, but the work looks like a student's, in which the facts are given without explanations and ideas about the mechanisms and structure. In this form, it cannot be published in the journal Molecules.
Specific remarks:
1. What interactions occur during synthesis? What active centers are then formed on the functionalized surface?
2. On which UV-Vis device and at what wavelength did you work?
3. In the experimental part, there are no methods and devices on which the results are obtained XRD patterns, IR spectra, SEM.
4. The data obtained in Fig. 1 and Fig. 2 are very poorly described. A couple of bands or peaks do not give information about what kind of new material you have. Refer to the literature once again and analyze the obtained data. Give an idea of the surface structure of the original and modified materials.
5. Why don't you analyze the appearance of P on the EDX spectrum in Fig. 3F?
6. Lines 229-232 - the description of the desorption experiment should be in the experimental part
This work is devoted to the removal of benzothiophene from heptane solutions using bioadsorbents. The topic is interesting, but the work looks like a student's, in which the facts are given without explanations and ideas about the mechanisms and structure. In this form, it cannot be published in the journal Molecules.
Specific remarks:
1. What interactions occur during synthesis? What active centers are then formed on the functionalized surface?
2. On which UV-Vis device and at what wavelength did you work?
3. In the experimental part, there are no methods and devices on which the results are obtained XRD patterns, IR spectra, SEM.
4. The data obtained in Fig. 1 and Fig. 2 are very poorly described. A couple of bands or peaks do not give information about what kind of new material you have. Refer to the literature once again and analyze the obtained data. Give an idea of the surface structure of the original and modified materials.
5. Why don't you analyze the appearance of P on the EDX spectrum in Fig. 3F?
6. Lines 229-232 - the description of the desorption experiment should be in the experimental part
Author Response
Thanks for your time and efforts to evaluate the manuscript comments which will improve the final version of the manuscript. We have revised the manuscript according to comments and highlight the changes with yellow background.

Round 2
Reviewer 1 Report
The authors have carefully revised and are currently acceptable.
Reviewer 2 Report
Accept in present form